# The Polyunsaturated Fatty Acid EPA, but Not DHA, Enhances Neurotrophic Factor Expression through Epigenetic Mechanisms and Protects against Parkinsonian Neuronal Cell Death

**DOI:** 10.3390/ijms232416176

**Published:** 2022-12-19

**Authors:** Maria Rachele Ceccarini, Veronica Ceccarelli, Michela Codini, Katia Fettucciari, Mario Calvitti, Samuela Cataldi, Elisabetta Albi, Alba Vecchini, Tommaso Beccari

**Affiliations:** 1Department of Pharmaceutical Sciences, University of Perugia, 06132 Perugia, Italy; 2Department of Medicine and Surgery, University of Perugia, P.le L. Severi, 1, 06132 Perugia, Italy

**Keywords:** PUFAs, EPA, DHA, Parkinson’s disease, neurodegeneration, apoptosis, mitochondrial damage, DNA methylation, BDNF, GDNF

## Abstract

ω-3 Polyunsaturated fatty acids (PUFAs) have been found to exert many actions, including neuroprotective effects. In this regard, the exact molecular mechanisms are not well understood. Parkinson’s disease (PD) is the second most common age-related neurodegenerative disease. Emerging evidence supports the hypothesis that PD is the result of complex interactions between genetic abnormalities, environmental toxins, mitochondrial dysfunction, and other cellular processes, such as DNA methylation. In this context, BDNF (brain-derived neurotrophic factor) and GDNF (glial cell line-derived neurotrophic factor) have a pivotal role because they are both involved in neuron differentiation, survival, and synaptogenesis. In this study, we aimed to elucidate the potential role of two PUFAs, eicosapentaenoic acid (EPA) and docosahexaenoic acid (DHA), and their effects on BDNF and GDNF expression in the SH-SY5Y cell line. Cell viability was determined using the MTT assay, and flow cytometry analysis was used to verify the level of apoptosis. Transmission electron microscopy was performed to observe the cell ultrastructure and mitochondria morphology. BDNF and GDNF protein levels and mRNA were assayed by Western blotting and RT-PCR, respectively. Finally, methylated and hydroxymethylated DNA immunoprecipitation were performed in the BDNF and GDNF promoter regions. EPA, but not DHA, is able (i) to reduce the neurotoxic effect of neurotoxin 6-hydroxydopamine (6-OHDA) in vitro, (ii) to re-establish mitochondrial function, and (iii) to increase BNDF and GDNF expression via epigenetic mechanisms.

## 1. Introduction

ω-3 Polyunsaturated fatty acids (PUFAs) are members of a large group of fatty acids (FA) possessing double bonds in their structure [1]. EPA (eicosapentaenoic acid; 20:5 ω-3) and DHA (docosahexaenoic acid; 22:6 ω-3) are very long-chain PUFAs that occur naturally in the highest quantities in fish [2]. In mammals, ω-3 PUFAs cannot be synthesized endogenously; therefore, the predominant source of EPA and DHA is dietary [3] and in a special kind of seafood.

ω-3 PUFAs have well-established anti-inflammatory properties [4] and have found clinical utility for cardiovascular disease prophylaxis and severe hypertriglyceridemia [5], with emerging evidence that they may be beneficial for the treatment of inflammatory diseases [6]. Recently, Tatsumi and co-workers reported that DHA and EPA induced antioxidant enzymes and protected Schwann cells from oxidative stress and, consequently, from diabetic neuropathy [7,8].

The dietary intake of ω-3 PUFAs has also been shown in human studies to inversely correlate with the overall risk of different types of cancer [9,10,11,12]. A substantial number of in vitro studies on cancer cell lines, as well as studies on animal models of cancers, have shown the antiproliferative, apoptotic, angiogenesis, cytotoxic, and antimetastatic properties of DHA and EPA [13,14,15]. Recent evidence also points to the potent and, at the same time, selective actions of EPA and DHA on multiple types of cancer cells [16]. Although ample data are available regarding the effects responsible for their actions on cancer, the exact mechanism is still ambiguous; however, recently, it was demonstrated that new mechanisms may involve ω-3 PUFAs as modulators of the aberrant epigenetic assembly in cancer cells [17,18].

PUFAs have become an emerging dietary medical intervention for health maintenance [19,20] and the treatment of neurodegenerative diseases such as Alzheimer’s disease (AD) and Parkinson’s disease (PD) [21]. Although their mechanism remains unclear, PUFAs seem to play a pivotal role in neural functions due to their abundance in neural tissues. Arachidonic acid (AA) and DHA are the two most abundant PUFAs within the nervous system and together comprise roughly 35% of the lipid content in the brain tissue [22].

PD is a chronic neurodegenerative disorder characterized by the progressive loss of dopaminergic (DA) neurons in the substantia nigra (SN), a subsequent decrease in dopamine in the striatum, and the presence of Lewy bodies (LBs), in which specific proteins, including α-synuclein, are deposited [23]. Dopaminergic degeneration is highly linked to the lack of neurotrophic factors, i.e., brain-derived neurotrophic factor (BDNF) and glial cell line-derived neurotrophic factor (GDNF), in neurons or the brain associated with PD [24]. The neurochemical features of PD include reactive oxygen species (ROS) generation, mitochondrial dysfunction, inflammation, the accumulation of misfolded proteins, nitric oxide production, and ubiquitin-proteasome system dysfunction [25]. Aging, as well as environmental and genetic factors, are considered disease risk factors for PD [25], whose symptoms include movement disorders such as resting tremors, postural abnormalities, rigidity, and akinesia, all of which develop as a result of the loss of 50–70% of DA neurons [26]. Despite its high incidence today, current pharmacotherapy for PD is lacking, and can only partially tackle symptoms of the disorder, but it is still unable to stop the progressive degeneration of neurons [27].

In addition to genetic changes, epigenetic mechanisms have been found to contribute significantly to PD [28,29]. The level of DNA methylation is one of the most studied epigenetic modifications in PD [30,31]. Epigenetic factors in general are chemical modifications of chromatin that do not change the underlying genomic sequence. These modifications can modulate gene expression, allowing differentiation into peculiar cellular phenotypes by driving tissue-specific expression patterns [32]. In particular, DNA methylation involves the conversion of cytosine to 5′-methyl cytosine (5mC) by at least five types of DNA methyltransferases (DNMTs), and PUFAs take part in this pathway [33]. Such methylation occurs in so-called “CpG islands” (cytosine-phosphate-guanine), which are regions of DNA rich in the dinucleotide formed by cytosine followed by guanine. This mechanism, together with the eventual deacetylation of histone lysine, is responsible for the compact conformation of chromatin, which inactivates the transcription of the genes of interest by preventing the access of transcription factors to promoter regions rich in CpG islands [34,35]. Discovered in 2009, DNA hydroxymethylation (5hmC) is a relatively new epigenetic modification occurring on cytosine [36] by the Ten-Eleven Translocation (TET) protein-mediated oxidative catalysis of 5mC [37,38]. While 5mC drives gene silencing, 5hmC plays some role in maintaining and/or promoting gene expression, but the exact mechanism is still to be elucidated [39].

Many PD-associated genes appear to be hypomethylated. α-Synuclein, one of the hallmark genes involved in PD, exhibits a hypomethylated island in intron 1, which determines a high expression level, whereas other genes are hypermethylated, such as MAPT (microtubule-associated protein tau), which exhibits a higher methylation level correlated with lower expression [31]. Consequently, the discovery of new compounds affecting DNMTs or TET expression might be promising tools for epigenetic therapy.

In this context, neurotrophic factors play a key role in the development, phenotypic maintenance, and survival of neurons in PD [40]. BDNF and GDNF are particularly relevant in this regard because they are involved in neuron differentiation and synaptogenesis [41,42]. Thus, the down-regulation of BDNF has been demonstrated in the SN in individuals with PD [43], but the reduced expression of GDNF in mice causes a marked reduction in DA neurons [44].

Elevated BDNF DNA methylation in exon IV down-regulates its expression in neurons [45], whereas hypermethylation in the promoter of GDNF influences its expression in brain tissue [46]. GDNF and BDNF expression levels may be, at least in part, under epigenetic control. As previously mentioned, PUFAs exert beneficial effects in the brain and could prevent or attenuate neurodegenerative diseases [47]. A correlation between a higher intake of ω-3 PUFAs from fish, rich in EPA and DHA, and a lower prevalence of PD was found [48,49]. They are able to reduce the risk of developing PD [50,51]. Some of the positive effects of ω-3 PUFAs on PD are probably due to their neuroprotective properties, since oxidative stress and neuroinflammation are ameliorated [21,52]. Apart from neuroprotective effects, ω-3 PUFAs may enhance DA signal transduction [53]. However, the role of each ω-3 PUFA has not been established, and in general, EPA and DHA are used as a mixture because their molecular mechanisms are not clear yet. Recently, we found that EPA modulates gene expression through epigenetic action in leukemia and in solid tumor cells [54,55].

In this study, we have analyzed the potential role of EPA and DHA, used individually, on cell viability, apoptosis, mitochondrial integrity, and BDNF and GDNF expression in the SH-SY5Y cell line. In this cell line, using neurotoxin 6-hydroxydopamine (6-OHDA), we mimicked cellular death and damage typical of neurodegenerative disorders, such as PD. The compound 5-aza-2′-deoxycytidine (5-AZA), an inhibitor of DNA methylation mainly employed in the treatment of several types of cancer, was used as a control for demethylation.

## 2. Results

### 2.1. Cell Viability by MTT with All Different Compounds Used Individually and in Combination

In our experiments, a neuronal cell line (SH-SY5Y) was used as an in vitro model. To mimic neurodegeneration, 6-OHDA, an analogue of the natural neurotransmitter dopamine, was utilized because its presence has been found in both rat and human brains [56]. Since its first description in 1959, 6-OHDA has played a fundamental role in preclinical research on PD [57]. First, to demonstrate the effects of 6-OHDA, EPA and DHA and their combination on the SH-SY5Y cell line, an MTT assay was performed after 24 h of treatment. For the above-mentioned test, a BSA/NaCl solution was used as a vehicle treatment because both EPA and DHA were prepared in this solution (Figure 1A). The concentrations used, from 1 to 20 µL, were safe. We observed a slight cytotoxic effect only after 50 µL of treatment. The toxin 6-OHDA diluted in PBS 1X, as previously reported [58,59], has a strong effect on cell viability. As shown in Figure 1B, SH-SY5Y viability decreased, and this cytotoxicity was dose-dependent. The first two concentrations, 10 µM (94.57 ± 3.26%) and 25 µM (87.35 ± 2.52%), had a moderate toxic effect. Significant cell death was observed with 100, 150, 200, and 500 µM. The dose chosen for inducing cell cytotoxicity and for evaluating the possible protective effects of EPA or DHA in all experiments was 50 µM 6-OHDA, which resulted in 71.37 ± 3.07% relative cell viability (*p* < 0.0001). Furthermore, 24 h of treatment with EPA (Figure 1C) and DHA (Figure 1D) alone from 10 up to 150 µM had no significant effect on cell viability, except for the two maximum doses tested, namely, 200 and 500 µM. As shown in Figure 1E,F, 6-OHDA was always present at a 50 µM concentration, whereas EPA and DHA were used from 10 to 500 µM. EPA incubation significantly prevented 6-OHDA-induced cell cytotoxicity; in particular, 100 µM EPA (Figure 1E) was able to return the cell viability approximatively to the control (CTR) value (95.09 ± 4.12%). DHA (Figure 1F) had a slight effect on cell viability, with best results at 50, 100, and 200 µM with viability around 87%. For this reason and because it was previously reported in the literature as the physiological dose [17], 100 µM of the two ω-3 PUFAs was selected as a standard dose in the following experiments.

### 2.2. Flow Cytometry Analysis Showed a Strong Induction of Apoptosis after 6-OHDA Treatment, Whereas EPA Had a Protective Effect

The apoptosis values were determined after 24 h in CTR and 6-OHDA-, EPA-, DHA-, and 5-AZA-treated cells, as shown in Figure 2, by flow cytometry. Using 50 µM 6-OHDA alone, we obtained a strong effect (*p* < 0.0001) on apoptosis (20.73 ± 2.48%) compared to untreated cells (1.43 ± 0.32%) (Figure 2), in accordance with MTT results. PUFA treatments alone (100 µM) did not induce SH-SY5Y apoptosis. Together with 6-OHDA, EPA treatment produced the greatest effect on reducing apoptosis (6.57 ± 1.37%) (Figure 2), but not DHA (17.37 ± 5.16%) (Figure 2). In addition, 5-AZA (1 µM) had a safe profile in terms of apoptosis (Figure 2), similar to PUFA treatments. The effects of all compounds on the percent distribution of the various phases of the cell cycle were studied, with no significant changes observed (data not shown).

Flow cytometry analysis was used before every experiment to ensure the correct cellular response.

### 2.3. Cell Morphology and Mitochondrial Organization Using Transmission Electron Microscopy (TEM) Analysis Were Compromised after 6-OHDA Treatment, but EPA Was Able to Reduce Damage

Transmission electron microscopy (TEM) was performed to observe the cell ultrastructure and mitochondrial morphology. The 50 µM 6-OHDA treatment after 24 h of incubation in vitro caused the swelling of mitochondria and damage to the cristae in the SH-SY5Y model (Figure 3B) compared to untreated cells, where mitochondria showed normal morphology with preserved cristae (Figure 3A). Electron-dense cytoplasmic inclusions and vacuoles of various sizes were also observed after 6-OHDA treatment (Figure 3B). Neither 100 µM EPA (Figure 3C) nor 100 µM DHA treatment (Figure 3E) altered the morphology of mitochondria, and no vacuoles or electron-dense cytoplasmic inclusions were observed. TEM showed that EPA reduces 6-OHDA-induced damage to mitochondrial membranes and cristae. No electron-dense cytoplasmic inclusions were found, and a few small cytoplasmic vacuoles were present (Figure 3D). DHA + 6-OHDA-treated cells showed partially swollen mitochondria, and small cytoplasmic vacuoles were visible in the cytoplasm (Figure 3F). This result shows that DHA has a less protective effect than EPA against 6-OHDA-induced toxicity. Treatment with 1 µM 5-AZA did not alter the morphology of mitochondria (Figure 3G).

### 2.4. BDNF and GDNF Protein Levels by Western Blot Analysis Increased Significantly after EPA Treatment

Western blotting was used to investigate the roles of 6-OHDA, EPA, DHA, and 5-AZA on BDNF and GDNF protein levels in the SH-SY5Y neuronal cell line. Treatment with 50 µM 6-OHDA induced reductions in BDNF and GDNF protein expression of 0.59- (Figure 4A) and 0.53-fold (Figure 4B), respectively, compared with CTR (arbitrarily set to 1). Significant increments in BDNF (*p* < 0.0001) and GDNF (*p* < 0.0001) were observed after 100 µM EPA treatment (2.42 and 1.73 times, respectively, in comparison with CTR, as shown in Figure 4). This ability of EPA to up-regulate BDNF and GDNF protein expression was also confirmed in combination with 50 µM 6-OHDA: 2.18 times for BDNF (*p* < 0.0001) and 1.66 times for GDNF (*p* < 0.0001) vs. CTR. As shown in Figure 4, different results were obtained with DHA. In fact, DHA alone was unable to significantly change the neurotrophic factor levels. The combination (100 µM DHA + 50 µM 6-OHDA) seems to slightly improve the effect of 6-OHDA alone, returning BDNF and GNDF expression to levels similar to CTR. The 1 µM 5-AZA treatment significantly increased BDNF (*p* < 0.01) and GDNF (*p* < 0.0001) protein levels (Figure 4).

### 2.5. BDNF and GDNF Genes Were Up-Regulated after EPA Treatment

To confirm the Western blotting data, BDNF and GDNF mRNA levels were determined by RT-PCR using the TaqMan Assay. SH-SY5Y cells were treated with the same doses of chemical compounds used for the previous experiments. Similar results were recorded with 6-OHDA, EPA, and DHA treatments in terms of gene expression in comparison with CTR (Figure 5). The BDNF mRNA level significantly increased (*p* < 0.0001) by 2.46 and 2.12 times, respectively, with EPA and EPA + 6-OHDA treatments (Figure 5A). On the contrary, DHA treatment did not change BDNF expression. A similar trend was obtained for the GDNF level (Figure 5B). EPA significantly increased (*p* < 0.0001) GDNF expression by 1.72 and 1.54 times, respectively, alone and in combination with 6-OHDA, whereas after DHA treatment, the GDNF level remained unchanged or even slightly decreased in combination with 6-OHDA (*p* < 0.01). 5-AZA was able to increase the BDNF and GDNF levels by 1.58 and 1.39 times, respectively (*p* < 0.001).

### 2.6. Different Effects of EPA and DHA on Hypermethylated CpG Islands Located in Promoter Regions of BDNF and GDNF Genes by DNA Immunoprecipitation (DNA IP)

Firstly, we sought CpG islands in the *GDNF* and *BDNF* promoter regions. In *BDNF*, a hypermethylated sequence (−4721/−5265) was identified; it is 400 bp long and contains 27 CpG. A hypermethylated region consisting of 712 bp was found in the *GDNF* promoter sequence (−7321/−8033) containing 59 CpG. To evaluate the methylation levels of these two CpG islands after 24 h EPA and DHA treatments, we performed a DNA IP assay using 5mC and 5hmC antibodies. The methylation and hydroxymethylation levels are approximatively the same in the control cells for both genes. Notably, we observed different effects in the treatment with the two PUFAs independently. In particular, the MeDIP assay demonstrated that 5mC levels decreased significantly (*p* < 0.0001) after 24 h EPA conditioning (Figure 6) in both promoter regions compared to CTR. Conversely, methylation in the *BDNF* promoter region (Figure 6A) remained almost stable, with a slight drop after DHA treatment compared to CTR, but decreased considerably (*p* < 0.0001) in the *GDNF* promoter region (Figure 6B). The hMeDIP assay revealed an important increase (*p* < 0.0001) in 5hmC content with 24 h EPA treatment, particularly for GDNF, showing the active transcription of both genes, in agreement with Western blot analysis and RT-PCR analysis data. The 5hmC levels did not change significantly after DHA conditioning in either gene (Figure 6) in comparison to untreated cells.

## 3. Discussion

This is the first study that reports a unique effect of EPA on a human neuronal cell line, that is, suppressing apoptosis via mitochondrial integrity and neurotrophin over-expression. In this context, several studies have shown the protective effects of dietary enrichment with PUFAs in different animal models of neurodegenerative diseases and clinical/epidemiological studies [60,61]. These studies showed that the prevalence of AD and PD negatively correlates with fish consumption. EPA has a central role in neurodegenerative disorders [62] but right now is used in combination with other PUFAs, particularly with DHA, because the exact molecular mechanisms are still not elucidated.

The aim of the present study was to understand which PUFAs (EPA or DHA) are able to prevent cell apoptosis, to reduce mitochondrial damage, and to re-establish the abnormal epigenetic modifications involved in the pathogenesis of neurodegenerative disorders, particularly PD.

Neurotoxin 6-OHDA, an analogue of the natural neurotransmitter dopamine, has a dose-dependent toxicity [58,59] and causes mitochondrial dysfunction [63] mimicking neurodegenerative disorders, such as PD [57]. To investigate this mitochondrial dysfunction, we used transmission electron microscopy on SH-SY5Y cells treated with 50 µM 6-OHDA. Treatment with 50 µM 6-OHDA caused the swelling of mitochondria and damage to the cristae, whereas PUFAs were safe, and treated cells looked like CTR cells. In this regard, it was recently demonstrated that the toxic effects induced by the herbicide Paraquat (1,1′-dimethyl-4-4′-bipyridinium dichloride) in Drosophila melanogaster were prevented by the concomitant dietary ingestion of an EPA/DHA mixture, and more interestingly, this neuroprotective effect involves mitochondrial protection [64]. In our study, we demonstrated, for the first time, that EPA was able to reduce 6-OHDA-induced damage to mitochondrial membranes and cristae better than DHA. In fact, TEM images indicate that EPA had a protective role against mitochondria swelling and damage to the cristae.

Recent studies suggest that neurodegeneration is due to abnormal DNA methylation, one of the most studied epigenetic modifications in PD [31,65,66]. Epigenetic factors do not change the underlying genomic sequence but may modulate gene expression. In this study, using the DNA IP technique, we were able to discriminate 5mC and 5hmC in two crucial neurotrophic factors, *BDNF* and *GDNF* genes, both important for the survival, maintenance, and regeneration of specific neuronal populations in the adult brain [67].

Our results show that EPA but not DHA is able to modify the methylation and hydroxymethylation levels of *BDNF* and *GDNF* and, as a consequence, their mRNA levels and protein expression. In particular, EPA considerably decreased 5mC and increased 5hmC in the promoter regions of these two important genes, crucial for neuron differentiation and synaptogenesis. The first one, a member of the neurotrophin family of growth factors, has been shown to modulate the survival, differentiation, synaptic plasticity, and activity of neurons. In PD, a reduction in BDNF expression in SN might contribute to the death of DA neurons because inhibiting BDNF expression in the SN causes parkinsonism in the rat [68]. A similar role was supposed for GDNF, but recent preclinical data suggest that GDNF can rescue DA neurons, and a possible therapy was evaluated for PD patients in clinical trials [69].

Replacement strategies and the administration of a therapy involving these neurotrophic proteins were considered in the past as potential therapeutics for PD [67], but the major problems found are the blood–brain barrier and other secondary side effects, such as anorexia and weight loss [70]. So, novel therapeutic approaches are needed. In this scenario, Nam and co-workers demonstrated that the human ras homolog enriched in the brain, which has an S16H mutation [hRheb(S16H)], protects the nigrostriatal DA projection in the 6-OHDA-treated animal PD model thanks to BDNF and GDNF activation [71]. Animal studies also revealed that physical activity promotes angiogenesis and neuroplasticity and may activate some extracellular signals, including BDNF [72].

Recently, Che and co-workers [73] demonstrated that EPA-enriched ethanolamine plasmalogen (EPA-pPE) enhanced BDNF/TrkB/CREB signaling and inhibited neuronal apoptosis in vitro and in vivo better than EPA-enriched phosphatidylethanolamine (EPA-PE). They proposed this molecule as a potential therapeutic agent in long-term AD therapy. Moreover, ethyl-eicosapentaenoate (E-EPA) can protect against motor impairments, neurodegeneration, and inflammation [74] and partially attenuated the striatal DA turnover [75] in a 1-methyl-4-phenyl-1,2,3,6-tetrahydropyridine (MPTP)-probenecid (MPTP-P) mouse model of PD.

Since in mammals, both DHA and EPA cannot be synthesized de novo, their plasma levels primarily reflect dietary intake and a small amount of transformation from the precursor alpha-linolenic acid (ALA, 18:3 ω-3) [76,77]. As previously described, PUFAs are essential for normal brain functioning, and their reduction in plasma has been associated with major depression and other neurological diseases [78,79,80].

In animal models, ω-3 PUFA deficiency induced in offspring via the manipulation of the rat maternal diet causes the marked over-expression of many genes that encode neurotransmitter receptors, particularly those for dopamine, glutamate, and acetylcholine [81].

In this context, Sublette and co-workers suggested that the dietary intake of PUFAs (both absolute and relative to the omega-6 status) may also contribute to the regulation of brain DA functioning in humans [82]. Potential mechanisms whereby ω-3 PUFAs may regulate dopamine include reducing cortical MAO-B activity, thereby decreasing the degradation of monoamines [83]; improving dopamine neuron survival [84]; and influencing lipid rafts [85].

A possibility may be that an endogenous molecule and/or a molecule derived from the diet is able to impart the important ability to continuously produce BDNF and GDNF as therapeutic agents against neurodegeneration in DA neurons. We already know that ω-3 PUFA deficiency can reduce the nigrostriatal system’s ability to maintain homeostasis under oxidative conditions, which may enhance the risk of PD. In particular, in rodents fed a diet deficient in ω-3 PUFAs, a dramatic decrease in brain PUFA contents has been observed with an important reduction in BDNF expression in SN [86].

Our findings suggest that EPA could be much more effective than DHA as an epigenetic modulator in preventing the loss of DA neurons via BDNF and GDNF expression. Our previous studies showed that only EPA wielded an important function in DNA demethylation in solid tumor cells [46,47]. For this reason, we aimed to investigate whether it could be a neuroprotective agent in the SH-SY5Y cell line through epigenetic mechanisms. In fact, it was previously reported that the entire molecular process behind DNA demethylation involved two stages: increased 5hmC levels, due to greater TET1 binding to DNA [17], and decreased DNA methyltransferase 1 (DNMT1) mRNA and protein levels [18].

In conclusion, the present study strengthens the evidence that EPA is able to reduce the neurotoxic effect induced by the neurotoxin 6-OHDA on the SH-SY5Y cell line via methylation reduction and the hydroxymethylation activation of two characteristic genes: BDNF and GDNF. Additional in vivo and clinical studies are needed to define its therapeutic use in humans. In any case, we speculate that EPA supplementation may prevent neurodegeneration, and it could be administered before clinical manifestations.

Future research directions should consider PUFA administration as a single treatment and not as a complex mixture, particularly for neurodegenerative disorder treatment.

## 4. Materials and Methods

### 4.1. Materials

The human bone marrow neuroblastoma SH-SY5Y cell line was purchased from Istituto Zooprofilattico Sperimentale della Lombardia e dell’Emilia Romagna ‘Bruno Ubertini’ (Brescia, Italy). RPMI 1640, L-glutamine, trypsin, and ethylenediaminetetraacetic acid disodium and tetrasodium salt (EDTA) were from Microtech Srl (Pozzuoli, NA, Italy).

Fetal Bovine Serum (FBS), penicillin–streptomycin, Master Mix TaqMan^®^Gene Expression and all primers for RT-PCR were from Thermo Fisher Scientific (Waltham, MA, USA). Antibiotics and Dulbecco’s phosphate-buffered saline pH 7.4 (PBS) were from Invitrogen Srl (Milan, Italy). Dimethyl sulfoxide (DMSO), ethanol, hydrochloric acid, sodium chloride, sodium deoxycholate, and SDS (Sodium Dodecyl Sulfate) were purchased from Carlo Erba (Reagenti Srl, Milan, Italy). 6-Hydroxydopamine hydrochloride (6-OHDA), eicosapentaenoic acid (20:5, ω-3; EPA), docosahexaenoic acid (22:6, ω-3; DHA), bovine serum albumin fraction V (BSA; fatty acid-free), 5-aza-2′-deoxycytidine, and monoclonal anti-β-tubulin III (neuronal) (T8578) were purchased from Sigma (Saint Louis, MO 63178 USA). Anti-BDNF (TA328615) rabbit polyclonal antibody was from Origene (Rockville, MD 20850), whereas anti-GDNF (ab176564) rabbit monoclonal antibody, was from Abcam (Cambridge, MA, USA). Anti-β-tubulin antibody (Sigma-Aldrich) was used to normalize. Monoclonal horseradish peroxidase-conjugated goat anti-rabbit secondary antibodies were from Santa Cruz Biotechnology (Bergheimer Str. 89-2, 69115 Heidelberg, Germany). Immunoreactive bands were visualized using the SuperSignal West Pico Chemiluminescent Substrate (ThermoFisher Scientific, Via Morolense 5, 03013 Ferentino FR, Italy). The EpiQuik Hydroxymethylated DNA Immunoprecipitation (hMeDIP/P-1038-24) Kit and Methylamp Methylated DNA Capture (MeDIP/P1015-24) Kit were from TEMA ricerca Srl (Via XXI Ottobre 1944, 11/240055 Castenaso, Bologna, Italy).

### 4.2. Albumin-Bound Fatty Acid

As previously described [87], EPA and DHA were prepared as stock solutions (10 mM). In brief, EPA and DHA were diluted in ethanol and precipitated by adding NaOH (final concentration: 0.25 M). The precipitate was dried under nitrogen, reconstituted with 0.9% (*w*/*v*) NaCl, and stirred at RT for 10 min with defatted BSA fraction V (fatty acid free) (final concentration: 10% *w*/*v*) in 0.15 M NaCl. Each solution was adjusted to pH 7.4 with NaOH and stored in the dark in aliquots in liquid nitrogen. The fatty acid/BSA molar ratio was 4:1 [17].

### 4.3. Cell Culture and Treatment

SH-SY5Y neuroblastoma cells, a subclone of the original SK-N-SH bone-marrow-derived cells, were grown in monolayer cultures with RPMI 1640 supplemented with 10% heat-inactivated FBS, 1 mmol/L sodium pyruvate, 2 mM L-glutamine, and antibiotics (100 U/mL penicillin and 100 μg/mL streptomycin). The cells were maintained in a cell incubator at 37 °C in a humidified atmosphere containing 5% CO_2_. When the cells reached 80–90% confluence, the routine culture medium was aspirated, and SH-SY5Y cells were washed with PBS 1X. The cells were then harvested by 0.05% trypsin in 0.02% Na_4_EDTA for 1–2 min at 37 °C and suspended in 1:8 supplemented growth medium to be maintained in the exponential growth phase. For experiments, the cells were counted using a trypan blue dye exclusion assay and seeded at a standard concentration for different assays.

For all experiments, the cells were cultured in the presence or absence of different substances: 6-OHDA, EPA, DHA, and 5-AZA. Stock solutions of 6-OHDA (2 mM) and 5-AZA (1 mM) were prepared before using in PBS 1X, whereas stock solutions of EPA (10 mM) and DHA (10 mM) were in BSA/NaCl stored in liquid nitrogen, as described in Section 4.2. The final concentrations for TEM, Western blotting, and RT-PCR analysis were 50 µM for 6-OHDA, 100 µM for EPA, 100 µM for DHA, and 1 µM for 5-AZA.

### 4.4. Viability by MTT Assay

Cellular viability was assessed by the reduction of MTT to formazan [88]. SH-SY5Y cells were seeded onto a 96-well plate at a density of 1 × 10^4^ cells/well with RPMI 1640 complete medium. After 24 h, fresh complete medium was replaced for treatment with different concentrations (from 10 to 500 µM) of 6-OHDA, EPA, and DHA for 24 h. Then, MTT reagent was dissolved in PBS 1X and added to the culture at a 0.5 mg/mL final concentration. After 3 h incubation at 37 °C, the supernatant was carefully removed, and formazan salt crystals were dissolved in 200 μL of DMSO added to each well [89]. The absorbance (OD) values were measured spectrophotometrically at 540 nm using an automatic microplate reader (Eliza MAT 2000, DRG Instruments, GmbH). Each experiment was performed twice in quadruplicate, and cell viability was expressed as a relative percentage, as previously described [90,91].

### 4.5. Flow Cytometry Analysis of Apoptosis and Cell Cycle

SH-SY5Y cells treated for 24 h with 50 µM 6-OHDA, 100 μM fatty acids (EPA and DHA), and 1 μM 5-aza were recovered, washed, and centrifuged at 200× *g*, then the cell pellets were resuspended in 1 mL of hypotonic propidium iodide (PI) solution (50 μg/mL in 1% sodium citrate plus 0.1% Triton X-100; Sigma), and the samples were incubated at 4 °C in the dark for 2 h. The fluorescence of PI of single nuclei was evaluated with an EPICS XL-MCL flow cytometer (Beckman Coulter, Brea, CA, USA), and the data were processed with an Intercomp computer and analyzed with EXPO32 software (Beckman Coulter) [92,93,94]. The percentage of apoptosis (hypodiploid DNA content) was determined with EXPO32 software (Beckman Coulter) as previously described [93,94,95]. The cell cycle was examined by measuring DNA-bound PI fluorescence in the orange-red fluorescence channel (FL2) with linear amplification. The percentage of cells in each cell-cycle phase was evaluated with ModFit software (Verity Software House, Topsham, ME, USA) [96].

Flow cytometry analyses were repeated three times in independent experiments. DNA fluorescence flow cytometric profiles of one experiment are representative of three, and graphs showing the mean ± standard deviation of percentage hypodiploid obtained in three different experiments are shown. The data were analyzed as described in the Statistical Analysis section.

### 4.6. Transmission Electron Microscopy (TEM) Analysis

Cells were fixed with 2% glutaraldehyde in 0.1 M phosphate buffer for 2 h at 20 °C. The fixative was washed off twice with PBS and post-fixed with 1% OsO4 in PBS for 1 h.

Samples were dehydrated through ethanol gradients and embedded in Epon.

Ultrathin sections were cut with a diamond knife on a Reichert ultramicrotome; the sections were mounted on Formvar single-hole grids or 150-mesh grids, stained with uranyl acetate and lead citrate, and then examined at 80 kV under a Philips EM 208 microscope equipped with a digital camera (University Centre for Electron Microscopy, CUME, Perugia, Italy).

### 4.7. Western Blotting

After treatment, SH-SY5Y cells were harvested and lysed in ice-cold RIPA lysis buffer containing 150 mM sodium chloride, 1% NP-40, 0.5% sodium deoxycholate, 0.1% SDS (Sodium Dodecyl Sulfate), and 50 mM Tris pH 8.0. Protein concentration was determined according to the method of Bradford [97]. About 60 µg of protein was used to perform 10% SDS–polyacrylamide gel electrophoresis at 200 V for 1 h, as previously described [98]. Proteins were transferred onto 0.45 µm cellulose nitrate strip membranes (Sartorius Stedim Biotech S.A.) in transfer buffer for 1 h at 100 V at 4 °C. Membranes were blocked with 5% (*w*/*v*) non-fat dry milk in PBS, pH 7.5, for 1 h at RT. The blot was incubated overnight at 4 °C with anti-BDNF, anti-GDNF, and anti-β-tubulin III specific antibodies (1:1000). The blot was treated with horseradish peroxidase-conjugated goat anti-rabbit secondary antibodies (1:5000). SuperSignal West Pico Chemiluminescent Substrate (ThermoFisher Scientific) was used to detect the chemiluminescent (ECL) HRP substrate. The apparent molecular weights of BDNF and GDNF were calculated according to the migration of molecular size standards. Images were acquired using the VersaDoc Imaging System (Bio-Rad, Hercules, CA, USA), and signals were quantified using Quantity One software (Bio-Rad). The area densities of the bands were evaluated by densitometry scanning and analyzed with Scion Image.

### 4.8. Reverse Transcription Quantitative PCR (RTq-PCR)

SH-SY5Y cells were used for total RNA extraction, which was performed by using the RNAqueous ^®^-4PCR kit (Ambion Inc., Austin, TX, USA) as previously reported [99]. Before cDNA synthesis, the integrity of RNA was evaluated by denaturing electrophoresis in TAE 1.2% agarose gel. cDNA was synthesized using 1 μg of total RNA for all samples using a High-Capacity cDNA Reverse Transcription kit (Applied Biosystems, Foster City, CA, USA) under the following conditions: 50 °C for 2 min, 95 °C for 10 min, 95 °C for 15 s, and 60 °C for 1 min for 40 cycles. RTq-PCR was performed using Master Mix TaqMan^®^Gene Expression and the 7.500 RT-PCR instrument (Applied Biosystems), targeting genes in the Taqman Array 96-Well Plate (P/N: 4414292): brain-derived neurotrophic factor (*BDNF*, Hs02718934_s1), glial cell-derived neurotrophic factor (*GDNF*, Hs01931883_s1), and glyceraldehyde-3-phosphate dehydrogenase (*GAPDH*, Hs99999905_m1).

### 4.9. Methylated DNA Immunoprecipitation (MeDIP) and Hydroxymethylated DNA Immunoprecipitation (hMeDIP) Assays

Fragmented DNA (0.5 mg/sample; 200–1000 bp in size) was denatured for 10 min at 95 °C. Hydroxymethylated DNA immunoprecipitation (hMeDIP; P-1038-24; Epigentek) and methylated DNA immunoprecipitation (MeDIP; P-2019-24; Epigentek) assays were performed according to the manufacturer’s instructions.

Antibody buffer (100 µL) was added to the following antibodies: 1 µL of non-immune IgG (negative controls), 1 µL of anti-5hmC, and 1 µL of anti-5mC and incubated at room temperature for 60 min. After removing the antibody buffer from the wells and washing with diluted washing buffer, 0.5 µg of fragmented DNA was added to each well. After incubation for 90 min at room temperature on an orbital shaker, the reaction wells were washed with diluted washing buffer. Proteinase K was added for 15 min at 60 °C, and the reaction wells were incubated at 95 °C for 3 min to inactivate the enzyme.

DNA samples were subjected to qRT-PCR using Brilliant SYBR Green qPCR Master Mix.

The following primers were used to amplify the *BDNF* promoter region (−4721, −5265):

Forward, 5′-GGAAAGTTGTTGGGCTGGTT-3′;

Reverse, 5′-GTTTCCTAGGGCTGCCTTCT-3′.

The following primers were used to amplify the GDNF proximal promoter sequence (NG_011675):

Forward, 5′-TTCGGTGGTTTAGTGGGGTG-3′;

Reverse, 5′-CAGTGGAAAACTCCCTGCCT-3′.

Fold enrichment was calculated by means of the amplification efficiency ratio of hMeDIP or MeDIP DNA samples over non-immune IgG by using 2^Δ(IgG Ct−sample Ct)^ [17].

### 4.10. Statistical Analysis

Data from three independent experiments using at least four independent cultures per condition were analyzed by ANOVA. GraphPad Prism version 9.2.0.332 was used (GraphPad software, San Diego, CA, USA) for MTT, flow cytometry, RT-PCR, Western blotting, and immunoprecipitation analyses. Data are expressed as mean ± S.D. A one-way ANOVA test was performed, and the significance thresholds were set as * *p* < 0.01, ** *p* < 0.001, and *** *p* < 0.0001. The nonparametric Mann–Whitney test or Kruskal–Wallis with post hoc Dunn’s test was used.

## Figures and Tables

**Figure 1 ijms-23-16176-f001:**
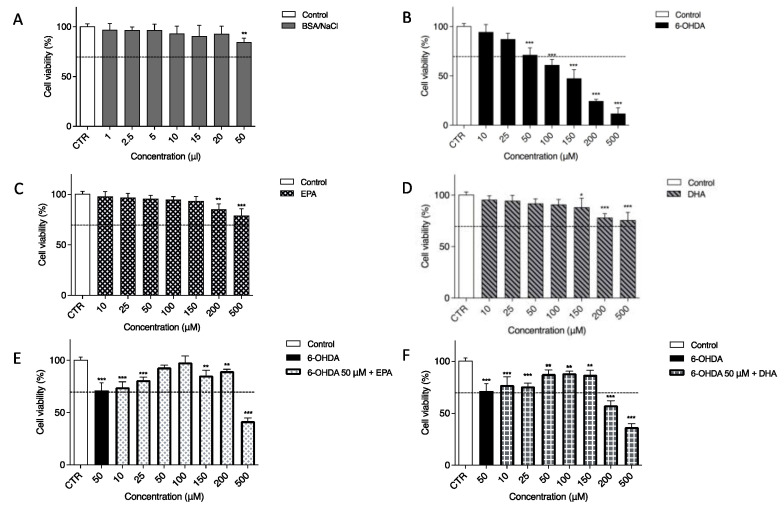
Effect of 6-OHDA, EPA, and DHA (alone or in combination) on the viability of SH-SY5Y cell line. Cells were treated for 24 h using BSA/NaCl solution as vehicle treatment condition (**A**) and different concentrations of 6-OHDA (**B**), EPA (**C**), or DHA (**D**). Based on this result, 50 μM 6-OHDA was used to treat the cells together with EPA (**E**) or DHA (**F**). In the last two cases, the first column represents untreated cells (white), the second column is the viability after 50 μM 6-OHDA treatment (black) and all other columns show co-treatment with 50 μM 6-OHDA and two omega-3 fatty acids. Cell viability was assessed using MTT assay. Data from three independent experiments using four independent cultures per condition were analyzed by ANOVA using Graphpad Prism software. * *p* < 0.01, ** *p* < 0.001, and *** *p* < 0.0001.

**Figure 2 ijms-23-16176-f002:**
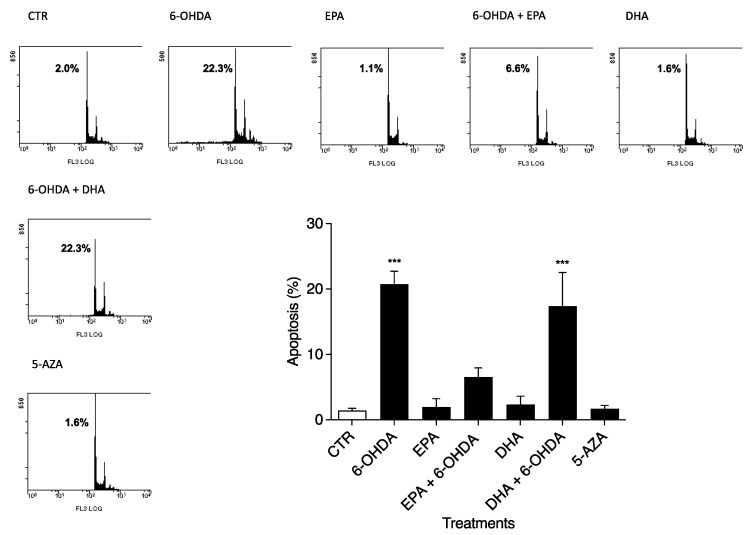
Effects of 6-OHDA, EPA, DHA, and 5-AZA on the apoptosis of SH-SY5Y cell line. Cells were treated for 24 h using 50 μM 6-OHDA, 100 μM EPA, 100 μM DHA, and 1 μM 5-AZA. Cells were harvested, stained with PI, and analyzed using flow cytometry. Representative plots are reported together with means ± S.D. Data from three independent experiments using four independent cultures per condition were analyzed by ANOVA using Graphpad Prism software *** *p* < 0.0001.

**Figure 3 ijms-23-16176-f003:**
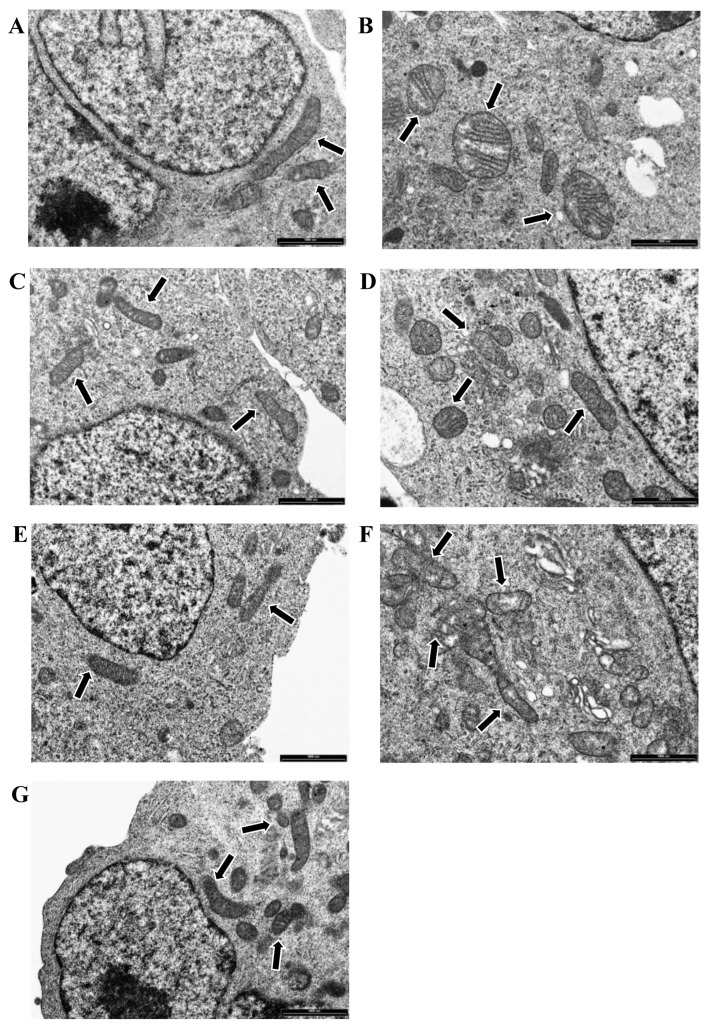
Representative transmission electron microscopy (TEM) images of CTR (**A**) and 6-OHDA- (**B**), EPA- (**C**), EPA + 6-OHDA- (**D**), DHA- (**E**), DHA + 6-OHDA- (**F**), and 5-AZA-treated (**G**) cells are reported. Mitochondria are highlighted with black arrows. Bars = 1000 nm.

**Figure 4 ijms-23-16176-f004:**
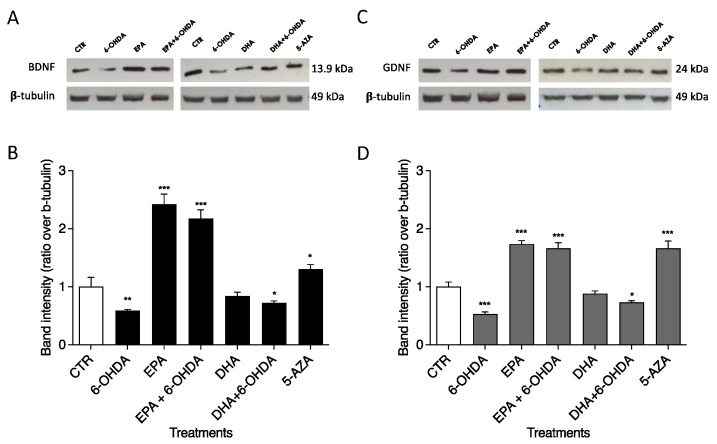
Effects of EPA and DHA alone and in combination with 6-OHDA on protein expression in SH-SY5Y cells. Cells were cultured and treated as reported in M&M. BDNF (**A**) and GDNF (**C**) levels were analyzed by Western blotting. The positions of proteins are indicated in relation to the positions of molecular size standards. β-Tubulin III (neuronal) was used to normalize protein levels. Area density was evaluated by Chemidoc Imagequant TL software. Values of BDNF (**B**) and GDNF (**D**) are expressed relative to the control, which was set to an arbitrary value (1.0). Experimental samples are reported as the mean ± SD. Data from three independent experiments using four independent cultures per condition were analyzed by ANOVA using Graphpad Prism software. * *p* < 0.01, ** *p* < 0.001, and *** *p* < 0.0001.

**Figure 5 ijms-23-16176-f005:**
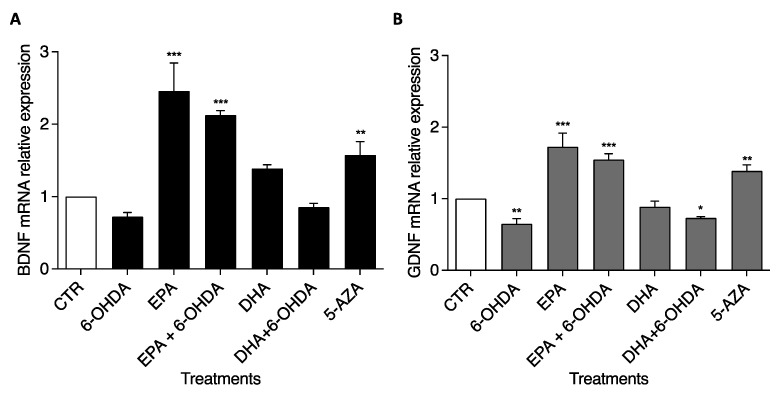
Effects of EPA and DHA alone and in combination with 6-OHDA on mRNA expression in SH-SY5Y cells. Cells were cultured and treated as reported in M&M. BDNF (**A**) and GDNF (**B**) gene expression levels were studied by RT-PCR. Glyceraldehyde 3-phosphate dehydrogenase (GAPDH) was used as a housekeeping gene. Experimental samples are reported as the mean ± SD. Data from three independent experiments using four independent cultures per condition were analyzed by ANOVA using Graphpad Prism software. * *p* < 0.01, ** *p* < 0.001, and *** *p* < 0.0001.

**Figure 6 ijms-23-16176-f006:**
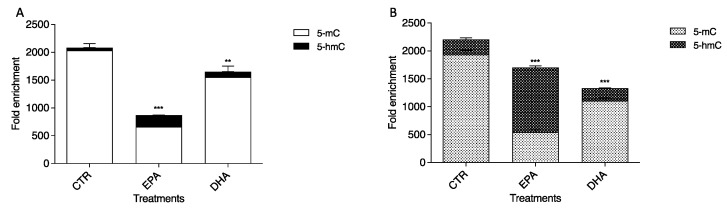
Effects of EPA and DHA on *BDNF* (**A**) and *GDNF* (**B**) methylation and hydroxymethylation levels. Experimental samples are reported as the mean ± SD. Data from three independent experiments using four independent cultures per condition were analyzed by ANOVA using Graphpad Prism software. ** *p* < 0.001, and *** *p* < 0.0001.

## Data Availability

The data that support the findings of this study are available from the corresponding author: Tommaso Beccari, email tommaso.beccari@unipg.it.

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
