# Peer review of "The Polyunsaturated Fatty Acid EPA, but Not DHA, Enhances Neurotrophic Factor Expression through Epigenetic Mechanisms and Protects against Parkinsonian Neuronal Cell Death"

_ijms, 2022, doi:10.3390/ijms232416176_

Round 1
Reviewer 1 Report
The work of Ceccarini and colleagues raises important points suggesting that there exist potential differences in the protective effects of EPA over DHA and they also explain how they believe this occurred and how it may be relevant for Parkinson’s therapies. While the work is interesting and may eventually provide valuable insights, there are major concerns that must be addressed.
Key Points
1. The most important of these is that a Vehicle treatment condition should have been used in all experiments rather than just comparing treatment data to untreated cells. For instance (1) BSA/NaCl solution should be used as a control for EPA or DHA and associated treatments, (2) the baseline solution in which 6-OHDA was prepared should be used as the control for 6-OHDA treatments. Using the same pH is also important and this is because components in Vehicle can have real effects. The good news is that such effects can be teased out statistically in the final analysis.
2. The authors disregarded the 200 uM treatments that were toxic based on propidium iodide cell counts and bar graphs. This part of the paper needs to be reconsidered after using Vehicle treatment as control. Also, as no cell cycle data were shown, this part of the results should be removed.
3. It is problematic to use any abbreviation before it is first defined. For instance in the title, EPA and DHA were used without telling readers they are polyunsaturated fatty acids. It might be allowable in the title if the authors at least tell the reader that they are polyunsaturated fatty acids. Also, authors had not yet defined BDNF or GDNF and based on findings, the authors seem to believe their data are relevant to Parkinson’s. Thus, a better and more accurate title would be: The polyunsaturated fatty acid EPA, but not DHA, enhances neuronal trophic factor expression and protects against parkinsonian neuronal cell death.
4. Many sentences start with a number or abbreviation, which is incorrect. In good English, sentences should start with a full word. For example, the authors could say “Using 6-OHDA” or “The toxin 6-OHDA” to accomplish this rather than starting a sentence with “6-OHDA”. The authors must also fix the many sentences that begin with:
a. n-3
b. 5hmC
c. 50 μM
d. any other number or abbreviation at the beginning of a sentence
5. The correct definition of the abbreviation GDNF is “glial cell line derived neurotrophic factor”, so this must be corrected on line 66 and elsewhere.
6. Regarding statistical analysis, authors must give more details regarding how they established that data showed significance. The number of replicates for each condition should be described in Methods as well as the version of software used not just a company. For instance … In legends the authors could state that “Data from three independent experiments using for example 3 - 5 independent cultures per condition analyzed by ANOVA using Prism software” then in methods state the vendor is Graphpad and tell where that company is located (San Diego, CA, USA).
7. The use of both "dopaminergic" and "DA" appear in the text. This should be uniform throughout but if you choose to use the abbreviation DA it must be defined before first use.
8. The SH-SY5Y neuroblastoma line is a sub-clone of the original SK-N-SH bone marrow derived cells, so please make this correction on line 352 of Methods. Also, SH-SY5Y cells are adrenergic, as they express dopamine beta hydroxylase, even though people use them as a dopaminergic model, it is technically incorrect to call them that.
9. In Figure 4, correct the symbol for beta on A and C rather than using the incorrect exclamation point (!) in front of tubulin.
10. Line 96, MAPTH must be corrected. Did you mean MAPT (microtubule associated protein tau)? Also as mentioned above, that abbreviation needs to be defined before using it.
11. Jellinger and colleagues confirmed 6-OHDA was measurable in human samples, however, it is incorrect to say 6-OHDA is "extensively" found in brain especially by just citing the Curtius reference, which is obscure, while Jellinger (1995) is accessible. It would also be worthwhile to explain these findings a little bit as many might not be aware. That way readers will learn that 6-OHDA can be measured in vivo and, thus it is a useful toxin for studying Parkinson's.
12. Line 72, rather than a review paper (reference 26), cite original literature which actually says that loss of 50-70% of striatal dopamine occurs after a large loss of nigral neuron and before parkinsonian symptoms appear.
13. For improved English usage, the following modifications of the Abstract should include: On line 12, authors began the sentence with n-3, but should begin with a full word. Also on line 12 "process" should be corrected to "possess". Line 13 "To this regard" should be "In this regard". Line 15 "evidences" should be "evidence", as that is already plural. Line 17 says "In this contest" but should be "context" also on line 100. Line 18, GDNF description is incorrect and should say "glial cell line derived neurotrophic factor". Line 20, needs to add the word "the" before "SH-SY5Y". Line 21, should remove the word "The" in front of then capitalize "Cell viability". Line 22, the word "progression" should be added at the end of the sentence. [However, as the authors never showed the data, that should be removed.] Line 23 change "expressions" to "protein levels and mRNA, as assessed by Western blot and RT-PCR. Line 26, change the word "damage" to "function" as you probably do not mean to say EPA caused mitochondrial damage.
Line 127, says "compuonds" ... please correct.
Line 142, incorrectly says 200 uM had no significant effect. It may not have been the dose that was selected, but it did have effects.
Lines 168-171 ...sinde no data are presented, take this out of the paper.
In section 2.3 the authors used the term "ridges" but "cristae" is the correct terminology for describing mitochondrial appearance and structure.
Line 213, rather than saying "abovementioned n-3 PUFA", just say DHA to be more clear.
Figure 5 needs to state which mRNA was actually evaluated on the Y axis of A and B.
Line 379, did you mean -220 C as with liquid nitrogen, as plus 220 C would not work? If so, just say "stored in aliquots in liquid nitrogen".
Section 4.3 needs to describe "recipes" for vehicle solutions for all treatment conditions.
Line 502, authors should reconsider as you wish to make data available once the paper is accepted so the research community will be more certain about the findings.
Author Response
Dear Editor,
Please see the attachment for all answers to Reviewer's requests.
English language and style have been extensively corrected by a colleague native speaking English.

Reviewer 2 Report
The manuscript entitled “EPA, but not DHA, promotes BDNF and GDNF expression throughout epigenetics mechanisms and protects neuronal cells from apoptosis” by Ceccarini and co-workers investigated the role of two PUFAs: EPA and DHA and their effects on BDNF and GDNF expression in SH-SY5Y cell line.
The authors analyzed the potential role of EPA and DHA using SH-SY5Y cell line. They assessed cell viability, apoptosis, mitochondrial integrity and BDNF and GDNF expression using neurotoxin 6-hydroxydopamine (6-OHDA) mimicking cellular death and damage typical in neurodegenerative disorders, such as Parkinson disease. The authors reported that EPA, and not DHA, is able to reduce neurotoxic effect of 6-OHDA in vitro, re-establish mitochondrial damage, and increase BNDF and GDNF expression by epigenetic mechanisms.
On the whole, this study seems well conducted, methods are sound, data are convincing and the conclusions are in general supported by the data. However, the discussion is short and leaves a certain amount of dissatisfaction.
A few issues, which need to be addressed:
· the abbreviation of 6-OHDA should be developed in the abstract
· the abbreviation of DA should be developed in the discussion
· in lines:128-129 in vitro should be in italics
Author Response
Dear Editor,
Please see the attachment for all answers to Reviewer's requests.
Minor spell checked have been correct by a colleague native speaking English.
